# Synergistic Inhibitory Effect of Multiple Polyphenols from Spice on Acrolein during High-Temperature Processing

**DOI:** 10.3390/foods12122326

**Published:** 2023-06-09

**Authors:** Juan Liu, Yongling Lu, Bo Si, Anqi Tong, Yang Lu, Lishuang Lv

**Affiliations:** 1Department of Food Science and Technology, School of Food Science and Pharmaceutical Engineering, Nanjing Normal University, 2 Xuelin Road, Nanjing 210023, China; 28190124@njnu.edu.cn (J.L.); 45227@njnu.edu.cn (Y.L.); 28190160@njnu.edu.cn (A.T.); luyang970516@163.com (Y.L.); 2National Liquor Product Quality Supervision and Inspection Center, Suqian Product Quality Supervision & Inspection Institute, 889 Fazhan Road, Suqian 223800, China; boston586@126.com

**Keywords:** acrolein, cardamonin, alpinetin, pinocembrin, *Alpinia katsumadai* Hayata, curcumin

## Abstract

Acrolein (ACR) is a toxic unsaturated aldehyde that is produced during food thermal processing. Here, we investigated the synergistic effect of polyphenols in binary, ternary, and quaternary combinations on ACR by the Chou–Talalay method, and then explored the synergistic effect of cardamonin (CAR), alpinetin (ALP), and pinocembrin (PIN) in fixed proportion from *Alpinia katsumadai* Hayata (AKH) combined with curcumin (CUR) in the model, and roasted pork using LC–MS/MS. Our results showed that their synergistic effect depended on the intensification of their individual trapping ACR activities, which resulted in the formation of more ACR adducts. In addition, by adding 1% AKH (as the carrier of CAR, ALP, and PIN) and 0.01% CUR (vs. 6% AKH single) as spices, more than 71.5% (vs. 54.0%) of ACR was eliminated in roast pork. Our results suggested that selective complex polyphenols can synergistically remove the toxic ACR that is produced in food processing.

## 1. Introduction

Acrolein (ACR) is an α,β-unsaturated aldehyde that is formed by amino acids, animal and vegetable fats, and carbohydrates [1], which is more toxic to cells than hydrogen peroxide (H_2_O_2_) and nearly equal in toxicity to hydroxyl radicals (•OH) [2]. It modifies proteins [3,4], DNA [5], RNA, and nucleotides and causes oxidative stress [6], mitochondrial dysfunction [7,8,9], and inflammatory reactions. Long-term exposure to ACR has been reported to increase the incidence of a variety of chronic diseases, including cardiovascular diseases, diabetes mellitus, cancer, and Alzheimer’s disease [10,11,12,13]. Food is one of the main sources of exogenous ACR *in vivo*, which can be produced by fat-rich foods under high-temperature cooking (stir-frying, frying, grilling, baking, etc.) [14,15], such as baked food (bread, cake, baked sausage, grilled steak, and grilled pork chops) [16,17,18,19,20], fried food (doughnuts, French chips, fried chicken, and fish) [21,22], and pickled food (salted pork) [23], and the highest ACR level in food can reach 11.1 mg/kg [24]. Currently, the World Health Organization has established the no-observed-effect level (NOEL) for ACR of 7.5 μg/kg body weight/day [25]. According to statistics, if only calculated according to the upper limit level of ACR in food, excluding environmental inhalation, the daily intake of ACR for adults in daily life is about 17 μg/kg body weight [26], far more than TDI. Another report revealed that there was a carcinogenic risk when the daily ACR exposure exceeds 50 μg/kg body weight [27]. Thus, reducing the amount of ACR in processed foods is one of the critical paths for lowering the level of ACR *in vivo*. Till now, various natural polyphenols have been well-documented to capture ACR through the formation of adducts to lower the amount of ACR, such as cyanidin-3-*O*-glucoside, myricetin, phloretin, luteolin, and resveratrol in vitro and *in vivo* [10,28,29,30,31]. Nevertheless, some polyphenols (myricetin, genistein, luteolin, etc.) are not authorized food additives and cannot be directly used in food, and some polyphenols (cyanidin-3-*O*-glucoside, phloretin, catechins, etc.) cannot endure high temperatures, which limits their application in high-temperature processed foods. Therefore, we are committed to screening thermostable ACR inhibitors from food additives or dietary polyphenols that can be used in food processing, singly or multiply.

Our previous research found that cardamonin (CAR) and alpinetin (ALP) could efficiently trap ACR under high-temperature conditions [32]. Because these two compounds are the major active components of *Alpinia katsumadai* Hayata (AKH) [33], AKH also had a certain ACR clearance effect, but the effect in our previous studies was not ideal. As a spice, AKH is a medicinal and edible food widely used alone or in combination with other spices to remove undesirable fishy or gamey odors and to enhance the aroma of meat processing. Generally, spices are often used in a variety of combinations in Chinese cooking, such as the most famous spice formula “The Thirteen Spices”, which results in the mixing of polyphenols and flavonoids in spices. How do multiple flavonoids and polyphenols cooperate to trap ACR when they coexist? As previously reported, curcumin in binary combination with quercetin could synergistically capture ACR in a roasted chicken wing [34]. Unfortunately, except for Japan, most countries do not allow quercetin to be used as a food additive. Therefore, more attention should be paid to whether other flavonoids or polyphenols in spices can also inhibit ACR in synergy with binary or multiple combinations, so as to provide constructive and guiding suggestions on how to use the combination of spices. 

In this study, our purpose was to screen a variety of polyphenols with the ability to capture ACR in high-temperature conditions, using a Chou–Talalay method to investigate the synergistic inhibitory effect of binary, ternary, and quaternary combinations of polyphenols on ACR, which could provide a scientific quantitative method for the determination of the combination index (CI) with n-compound combination, and the optimal combination ratio for the maximal synergic effect. Moreover, the synergistic inhibitory pathway was also explored through a quantitative analysis of ACR adduct-conjugated polyphenols using LC–MS/MS. In addition, we attempted to choose AKH as the carrier of three flavonoids further mixed with curcumin to explore the synergistic effect, and then apply it to real roasted pork processing. Our findings thus provide a strategy for enhancing the ACR inhibitory ability of polyphenols through multiple combinations. Moreover, a spice recipe that can not only give a rich flavor to food but can also be used as a promising new ACR inhibitor in food processing was developed.

## 2. Materials and Methods

### 2.1. Materials

ACR (AR, 95% in water) was purchased from Saen Chemical Technology Co., Ltd. (Shanghai, China). CAR (98%, HPLC) and pinocembrin (PIN, 98%, HPLC) were purchased from Xi’an Natural Field Bio Technique Co., Ltd. (Xi’an, China). ALP (98%, HPLC) was purchased from Chengdu Alfa Biotechnology Co., Ltd. (Chengdu, China). Quercetin (QUE, 98%, HPLC) and curcumin (CUR, 98%, HPLC) were purchased from Shanghai Macklin Biochemical Reagent Co., Ltd. (Shanghai, China). Kaempferol (KAE, 98%, HPLC) and galangin (GAL, 98%, HPLC) were purchased from Nanjing Guangrun Biotechnology Co., Ltd. (Nanjing, China). Rutin (RUT, 98%, HPLC) was purchased from Aladdin Reagent Co., Ltd. (Shanghai, China). CAR-ACR (92%, HPLC), ALP-ACR (98%, HPLC), and CUR-ACR (92%, HPLC) were prepared in our lab, and their structures were elucidated via NMR and HRMS. Liquid chromatography-grade solvents were obtained from Shanghai Sinopharm Chemical Reagent Co., Ltd. (Shanghai, China). AKH was purchased from Guiwei Wang Catering Management Co., Ltd. (Beijing, China). Streaky pork, salt, cooking wine, oyster sauce, and sugar were obtained from a local retail market (Suguo, Nanjing, China).

### 2.2. Determination of the Inhibitory Activities and IC_50_ of Polyphenols on ACR

The stock solution of different polyphenols was prepared in DMSO, and then diluted with PBS to a series of solutions for the determination. ACR (0.5 mmol/L) was incubated with CAR/PIN/CUR (0.125, 0.25, 0.5, 1.0, 1.5, and 2.0 mmol/L), ALP/KAE (0.25, 0.5, 1.0, 1.5, and 2.0 mmol/L), QUE (0.125, 0.25, 0.5, 1.0, and 1.5 mmol/L), GAL (0.25, 0.5, 1.0, 2.0, and 4.0 mmol/L) or RUT (0.125, 0.5, 1.5, and 2.0 mmol/L) in combination with PBS (0.1 mol/L, pH 7.0) at 100 °C for 60 min, and ACR incubated with PBS instead of polyphenols was used as a control sample. Then, 500 μL samples were mixed with 300 μL of 2,4-dinitrophenylhydrazine (DNPH) and continuously shaken for 2 h at 37 °C for derivatization. After that, the samples were analyzed for ACR-DNPH using LC-DAD based on our published methods [35]. Each sample was prepared in triplicate. Then, the inhibition rates of the polyphenols on ACR were calculated according to the equation. The IC_50_ values of all eight polyphenols were assayed using the web page https://www.aatbio.com/tools/ic50-calculator.
$$\begin{array}{l} \mathrm{Percentage}\;\mathrm{of}\;\mathrm{trapping}\;\mathrm{ACR}=\\ \frac{\mathrm{amount}\;\mathrm{of}\;\mathrm{ACR}\;\mathrm{in}\;\mathrm{control}\;(\mathrm{as}\;\mathrm{ACR}\;-\;\mathrm{DNPH})\;-\;\mathrm{amount}\;\mathrm{of}\;\mathrm{ACR}\;\mathrm{in}\;\mathrm{the}\;\mathrm{test}\;(\mathrm{as}\;\mathrm{ACR}\;-\;\mathrm{DNPH})}{\mathrm{amount}\;\mathrm{of}\;\mathrm{ACR}\;\mathrm{in}\;\mathrm{control}\;(\mathrm{as}\;\mathrm{ACR}\;-\;\mathrm{DNPH})}\times 100\% \end{array}$$

### 2.3. Inhibitory Effects on ACR of Polyphenols Individually or in Binary Combination That Were Incubated

ACR (0.5 mmol/L) was incubated with CAR (0.125, 0.25, 0.5, 1.0, 2.0, and 4.0 times IC_50_) and ALP/PIN/CUR/QUE/KAE/GAL/RUT (0.125, 0.25, 0.5, 1.0, 2.0 and 4.0 times IC_50_), individually or in combination in PBS (0.1 mol/L, pH 7.0) at 100 °C for 60 min, and ACR incubated with PBS instead of polyphenols was used as a control sample. To stop the reaction, each mixture was cooled in an ice bath. Then, the ACR-DNPH level was detected by LC-DAD using the above method. The Chou–Talalay equation was used to assay the inhibitory effect of CAR, in combination with seven other polyphenols, on trapping ACR using CompuSyn 2.0 software (ComboSyn, Inc., Paramus, NJ, USA).

### 2.4. Analysis of the Inhibitory Effect and Mechanism of CAR, ALP, PIN, and CUR in Quaternary Combination in Capturing ACR by Using LC–MS/MS

Stock solutions of CAR, ALP, PIN, and CUR with concentrations of 0.125 × IC_50_, 0.25 × IC_50_, 0.5 × IC_50_, 1.0 × IC_50_, 2.0 × IC_50_, and 4.0 × IC_50_ were prepared. ACR (0.5 mmol/L) was mixed with the above four polyphenols at various concentrations individually or in combination at 100 °C for 60 min, and ACR incubated with PBS instead of polyphenols was used as a control sample. An ice bath was used to stop the reaction. Then, the ACR-DNPH level and the inhibitory effect of the four polyphenols in quaternary combination were detected using the above method.

### 2.5. Inhibitory Effect on ACR of [CAR + ALP + PIN] at the Fixed Proportion in AKH and CUR in Quaternary Combination in the Model

Our quantitative analysis showed that the amounts of CAR, ALP, and PIN from AKH were 5.53, 3.30, and 8.27 mg/g, respectively (Appendix A). According to this ratio, the concentrations of the flavonoid combination were set at constant fold changes of 0.4, 0.5, 1.0, 2.0, and 4.0 times the concentrations of CAR (0.51 mmol/L), ALP (0.31 mmol/L), and PIN (0.81 mmol/L), and then the mixtures of the three compounds were incubated with ACR (0.5 mmol/L) at 100 °C for 1 h, and PBS, instead of three flavonoids, incubated with ACR was used as a control sample. An ice bath was used to stop the reaction. Then, the ACR-DNPH level and the inhibitory effect of the three flavonoids in combination were detected using the above method.

Furthermore, ACR (0.5 mmol/L) was incubated with a [CAR + ALP + PIN] mixture with a fixed ratio of 0.51:0.31:0.81 (0.0625, 0.125, 0.25, 0.5, 1.0, 2.0, and 4.0 times the IC_50_ values of the three flavonoids) and CUR (0.0625, 0.125, 0.25, 0.5, 1.0, 2.0 and 4.0 times IC_50_) individually or in combination in PBS (0.1 mol/L, pH 7.4) at 100 °C for 1 h, and ACR incubated with PBS instead of polyphenols was used as a control sample. At the end of the reaction, an ice bath was used to stop the reaction. Then, the ACR-DNPH level and the inhibitory effect of the four polyphenols in combination with ACR were evaluated using the above method.

### 2.6. Quantitative Analysis of the ACR Adducts of CAR, ALP, PIN, and CUR 

A series of standard solutions for the determination of CAR-ACR, ALP-ACR, and CUR-ACR [34,36] were prepared using the adducts (1.0 mg/mL) synthesized in our lab. Four standard curves were quantified with external standards (CAR-ACR: y = 535021x + 5813.33, R^2^ = 0.9981; ALP-ACR: y = 751878x + 3292.16, R^2^ = 0.9989; PIN: y = 643449x + 4168.89, R^2^ = 0.9964; CUR-ACR-1: y = 229561x + 3401.20, R^2^ = 0.9951). The range of linearity of the calibration curve was 0.04–2.00 μg/mL for CAR-ACR and ALP-ACR, 0.04–8.00 μg/mL for PIN, and 0.01–5.00 μg/mL for CUR-ACR-1. LOD and LOQ were calculated from the low concentration value of the calibration curves by considering the peak area corresponding to 3 (for LOD) or 10 (for LOQ) times the signal-to-noise ratio of a procedural blank estimated by the software of Waters. The limit of detection (LOD) and the limit of quantification (LOQ) were 0.0006 μg/mL and 0.0018 μg/mL for CAR-ACR, 0.0008 μg/mL and 0.0024 μg/mL for ALP-ACR, 0.0003 μg/mL and 0.009 μg/mL for PIN, and 0.0007 μg/mL and 0.0021 μg/mL for CUR-ACR-1, respectively. The ACR adducts of the CAR, ALP, PIN, and CUR were quantified according to the standard curves of CAR-ACR, ALP-ACR, PIN, and CUR-ACR-1.

### 2.7. ACR Inhibitory Activities of AKH and CUR Individually and in Combination in Roasted Pork

A pork belly sample was washed and cut into 5 × 5 × 3 cm squares and divided into eight groups (500 g of pork per group), namely, group A: blank control, group B: with the complex of [CAR + ALP + PIN] according to the fixed proportion of the three compounds in 1.0% AKH, group C: with AKH (1.0%), group D: with CUR (0.1%), group E: with [CAR + ALP + PIN] (1.0% AKH) + CUR (0.1%), and group F: with AKH (1.0%) + CUR (0.1%). Then, the six groups were all marinated with 3 g of salt, 3 g of light soy sauce, 3 g of oyster sauce, 5 g of cooking wine, and 5 g of cooking oil for 2 h. The sliced streaky pork was roasted in a CRTF32K oven (Changdi Electrical Technology Co., Ltd., Guangzhou, China) at 180 °C for 20 min (turned every 10 min). Samples were stored at −80 °C for further analysis.

After being lyophilized, the pork was homogenized with a mincer. A total of 3.0 g of each sample was removed and homogenized in 5.0 mL of water for 3 min before centrifugation. The supernatant was removed, and the precipitate was added to 5.0 mL of 50% aqueous methanol solution, followed by homogenization and centrifugation, and the supernatant was collected again. Both supernatants were combined and condensed. Then, the supernatants were derivatized by DNPH and detected using HPLC-DAD. Each sample was prepared in triplicate.

All samples (3.0 g) above were homogenized in 5 mL of acetonitrile-saturated n-hexane for 2 min and then homogenized in 15 mL of n-hexane-saturated acetonitrile for 2 min. After double extraction, the acetonitrile layer was combined and concentrated for drying, and methanol (1 mL) was used to dissolve the residue for LC–MS/MS analysis.

### 2.8. LC–MS/MS Analysis

In this study, an Agilent Mass Hunter system, consistent with our previous publication, was used [36].

For the analysis of CAR, ALP, PIN, and their adducts with ACR, a 250 × 4.6 mm i.d., 5 µm Eclipse XDB-C_18_ column (Agilent) was used with a flow rate of 0.6 mL/min. The mobile phases were water containing 0.1% formic acid (phase A) and acetonitrile (phase B). The column elution started with 60% A and 40% B for 3 min, which was linearly increased to 80% B from 3 to 15 min, maintained at 80% B from 15 to 18 min, and equilibrated to 60% A and 40% B from 18.1 to 21 min for the next run. The column was maintained at 30 °C. The detection wavelength was set to 300 nm. The LC eluent was introduced into the ESI interface. The positive ion polarity mode was set for the ESI ion source, with the voltage on the ESI interface maintained at approximately 4 kV. Nitrogen gas was used as the sheath gas at a flow rate of 45 arbitrary units and as the auxiliary gas at a flow rate of 5 arbitrary units. Structural information on CAR, ALP, PIN, and the major ACR adducts was obtained by tandem mass spectrometry (MS/MS) through collision-induced dissociation (CID).

For the analysis of CUR and its adducts with ACR, the same column and flow rate as above were used. The HPLC mobile phases were water containing 0.1% formic acid (phase A) and acetonitrile (phase B). The column elution started with 70% A and 30% B for 5 min, which was linearly increased to 50% B from 5 min to 8 min and to 90% B from 8 min to 13 min, subsequently decreased to 50% B from 13 min to 15 min, and finally decreased to 30% B from 15 min to 18 min. The detection wavelengths were set to 342 and 426 nm. The ESI conditions were the same as the above method, but set to negative mode. 

For the analysis of pork samples that were paired with AKH and CUR, the same column and flow rate as above were used. The mobile phases were water containing 0.1% formic acid (phase A) and acetonitrile (phase B). Gradient elution was performed using the following gradient: 30% B for 3 min, 30–75% B for 3–28 min, 75–100% B for 28–28.1 min, 100% B for 28.1–31 min, 100–30% B for 31–31.1 min, and 30% for 31.1–36 min for the next run. The injection volume was 20 μL, and the detection wavelengths were set to 300, 342, and 426 nm. 

### 2.9. Statistical Analysis

All results are expressed as the mean ± standard deviation. Statistical analyses were performed using GraphPad Prism. Tukey’s tests were used to compare significant differences among treatments. Each sample was analyzed in triplicate. *p* < 0.05 was considered significant. The synergism of polyphenols was analyzed with the Fa–CI plot, and CI calculations were performed according to the Chou–Talalay method using CompuSyn 2.0 software (ComboSyn, Inc., Paramus, NJ, USA). CI values below 1 suggest synergy, whereas CI values above 1 indicate antagonism.

## 3. Results and Discussion

### 3.1. Scavenging Capability of the Eight Polyphenols on ACR

The numbers of polyphenols rich in common spices were screened for the ability to inhibit ACR. As shown in Figure 1, all eight considered polyphenols (CAR, ALP, PIN, CUR, QUE, KAE, GAL, and RUT) inhibited ACR in a dose-dependent manner at a high temperature of 100 °C. Among them, CUR had an outstanding inhibitory activity on ACR, and up to 81.55% of ACR was eliminated with the addition of CUR (1.0 mmol/L), while RUT exhibited weak ACR trapping activity, and no more than 50% ACR was cleared at a concentration of 1.5 mmol/L. Meanwhile, the IC_50_ values of these inhibitors in response to ACR were recorded in descending order: GAL, RUT, KAE, ALP, CAR, PIN, CUR, and QUE. Here, we selected one hour to perform the following experiments because most foods are processed in less than an hour.

### 3.2. Scavenging Effects on ACR of Eight Polyphenols in Binary Combination

The Chou–Talalay method is usually used to define the type of drug-drug interaction. The combination index (CI) theorem offers a quantitative definition for the additive (CI = 1), synergistic (CI < 1), or antagonistic (CI > 1) effect of an inhibitor combination, and the smaller the CI value, the stronger the synergy [37]. According to the combination index theorem, our results showed that the combined treatment of CAR with CUR, ALP, or PIN (CI < 1) exhibited an obvious synergistic effect (Figure 2), while a stronger antagonistic effect was identified in the combination of CAR with QUE, GAL, KAE, or RUT (CI > 1, Appendix A).

As shown in Figure 2A, only the lower concentrations within 0.09–0.78 mmol/L CAR and 0.01–0.80 mmol/L ALP at a CAR to ALP ratio of 1:1.04 (Equivalent to IC_50_ ratio of two) resulted in a synergistic interaction between CAR and ALP. The calculated CI was approximately 0.80 for Fa above 0.55, which generated a more obvious effect than in the individual treatments.

As shown in Figure 2B, except for the highest concentration, within 0.09–2.56 mmol/L CAR and 0.03–2.26 mmol/L PIN, at a CAR to PIN ratio of 1:0.85, the combined treatment had a synergistic inhibitory effect on ACR, and the calculated CIs were approximately 0.39–0.81. The maximum synergistic inhibitory rate (Fa) reached approximately 85%.

As shown in Figure 2C, within 0.09–2.80 mmol/L CAR and 0.06–1.76 mmol/L CUR, at a CAR to CUR ratio of 1:0.62, a significant synergistic inhibitory effect on ACR was observed by CAR combined with CUR. The calculated CIs were less than 1 for all the studied combinations, while the majority of the CIs were approximately 0.50, and the maximum synergistic inhibitory rate (Fa) was nearly 100%.

### 3.3. Scavenging Effect on ACR of CAR, ALP, PIN, and CUR in Quaternary Combination

Considering the synergistic inhibitory effect of CAR in binary combination with ALP/PIN/CUR on ACR, we are more interested in the clearance effect on ACR of the four inhibitors (CAR, ALP, PIN, and CUR) in combination. As shown in Figure 2D, the four polyphenols in combination also exhibited synergistic inhibitory effects on ACR at very low concentrations, such as 0.09 mmol/L CAR, 0.10 mmol/L ALP, 0.08 mmol/L PIN, and 0.06 mmol/L CUR, and up to 52.31% ACR was eliminated, significantly higher than the single inhibitory rates of the four polyphenols at the same concentration (3.50%, 17.03%, 10.37%, and 11.46%, respectively). A particularly stronger synergistic effect was identified based on the calculated CIs, which were below 0.52 for all of the studied combinations.

### 3.4. Synergistic Scavenging Mechanism on ACR of the Quaternary Combination of CAR, ALP, PIN, and CUR 

To determine the synergistic inhibitory mechanism of the quaternary combination of CAR, ALP, PIN, and CUR, LC–MS/MS was used to analyze the incubation mixture with ACR. As shown in Figure 3, when CAR reacted with ACR at 100 °C for 10 min (Figure 3A), three new peaks appeared (Rt = 6.46, 8.01, and 15.38 min). The first new peak (Rt = 6.46 min) corresponded to a molecular ion [M + H]^+^ with *m*/*z* 271, which fragmented into *m*/*z* 167 [M + H]^+^, and the retention time and MS/MS fragments were the same as those of the ALP standard. Our previous studies showed that CAR could be transformed into ALP in the presence and absence of ACR at high temperatures [32]. The second and third new peaks (Rt = 8.01 and 15.38 min) corresponded to the same molecular ion [M + H]^+^ with *m*/*z* 327 (56 amu [Mw ACR, 56] more than ALP/CAR *m*/*z* 271 [M + H]^+^), and both of them were fragmented into *m*/*z* 283 [M − 44 + H]^+^ (loss of one [–CHOH–CH_2_–] group [*m*/*z* 44]) [29]. Their retention times and fragments were the same as those of the ALP-ACR (Rt = 8.01 min, mono-ACR adduct conjugated to ALP) and CAR-ACR standards (Appendix A) (Rt = 15.38 min, mono-ACR adduct conjugated to CAR) that were prepared in our lab [36]. Therefore, the three new peaks corresponded to ALP (Rt = 6.46 min), ALP-ACR (Rt = 8.01 min), and CAR-ACR (Rt = 15.38 min). The reaction path of CAR and ALP capturing ACR to form adducts is described in Figure 4a,b.

When ALP reacted with ACR at 100 °C (Figure 3B), an ALP-ACR peak (Rt = 8.01 min) appeared, and the retention time and the fragment of MS^2^ (Appendix A) were the same as those of the standard of ALP-ACR (mono-ACR adduct conjugated to ALP) that was prepared in our lab [32,36]. The reaction path of ALP capturing ACR to form an adduct is described in the Figure 4b. When PIN reacted with ACR at 100 °C for 10 min (Figure 3C), two new peaks appeared (Rt = 12.12 and 12.47 min) with the same molecular ion [M + H]^+^ with *m*/*z* 313 (56 amu [Mw ACR, 56] more than PIN *m*/*z* 257 [M + H]^+^) and the same fragment ions [M − 56 + H]^+^ with *m*/*z* 257.1 and [M − 44 + H]^+^ with *m*/*z* 269.1 (loss of one [–CHOH–CH_2_–] group [*m*/*z* 44]) (Appendix A) [38], thereby indicating that the two peaks were mono-ACR adducts that were conjugated to PIN, namely, PIN-ACR-1 (Rt = 12.12 min) and PIN-ACR-2 (Rt = 12.47 min), which were also found in mouse urine after gavage with CAR [36]. The reaction path of PIN capturing ACR to form adducts is described in the Figure 4c. When CUR reacted with ACR at 100 °C for 10 min (Figure 3D), at a wavelength of 342 nm, three new peaks appeared (Rt = 10.34, 11.58, and 12.08 min). The first and second new peaks corresponded to the same molecular ion [M − H]^−^ with *m*/*z* 423 and fragment ions [M − 56 − H]^−^ with *m*/*z* 367 and [M − 44 − H]^−^ with *m*/*z* 379 (Appendix A), and their retention times and MS^2^ fragments were similar to those of the CUR-ACR-1 and CUR-ACR-2 standards that were prepared in our laboratory [34]. The third new peak corresponded to the molecular ion [M − H]^−^ with *m*/*z* 479 and fragment ions [M − 56 − H]^−^ with *m*/*z* 423 and [M − 56 − 56 − H]^−^ with *m*/*z* 367 (Appendix A), which suggested the loss of one or two ACRs from CUR-2ACR, which consists of two molecular ACRs conjugated to CUR, similar to our previous study [34]. Moreover, at a wavelength of 426 nm, the content of CUR was significantly decreased, thereby suggesting that CUR transforms into the adducts of CUR that are conjugated with ACR (Figure 3D). The reaction path of CUR capturing ACR to form adducts is described in Figure 4d.

When CAR, ALP, PIN, and CUR were mixed and reacted with ACR, ACR adducts such as CAR-ACR, ALP-ACR, PIN-ACR-1, PIN-ACR-2, CUR-ACR-1, CUR-ACR-2, and CUR-2ACR were all detected (Figure 3E). Compared with each inhibitor used individually, the yield of corresponding ACR adducts formed by combination was remarkably increased, accompanied by a sharp decrease in CUR. Therefore, the synergistic inhibition mechanism that polyphenols capture more ACR leading to the formation of more ACR products is proposed.

### 3.5. Quantitative Analysis of the Synergistic Effect on ACR of Three Flavonoids (Fixed Proportion in AKH) and CUR in Combination in the Model

Our above result proves that an equimolar combination of CAR, ALP, PIN, and CUR within suitable concentration ranges synergistically inhibits ACR. However, CAR, ALP, and PIN are not food additives and have not been allowed to be used in food until now. Nevertheless, it is well documented that CAR, ALP, and PIN usually coexist in a fixed proportion in AKH [32], which is a commonly used additive in Chinese-style meat cooking and processing [39]; hence, we wonder if AKH can be used as a carrier to replace CAR, ALP, and PIN? If so, how do CAR, ALP, and PIN work together when executing a ternary combination based on a fixed proportion in AKH? Additionally, if the synergistic inhibition effect is good, is the added dose of AKH within a reasonable range for food processing? As a result, the optimum inhibitory rate of the ternary combination of [CAR + ALP + PIN] at a ratio of 5:3:8 (0.51:0.31:0.81 mmol/L, the native proportion of CAR, ALP, and PIN in AKH) [32] was investigated in order to find the suitable dosage of AKH. As shown in Appendix A, a desirable synergistic inhibitory effect of the three flavonoids was also observed when the equivalent dosage of AKH ranged from 4.27 to 100.03 g/kg, which is just within the dose ranges of commonly used spices in meat cooking and processing [40]. This obtained result also provided an effective dosage range for AKH to be used alone or in combination with other spices.

In order to further enhance the synergistic inhibitory effect in practical application, we further optimized the inhibitory effect of CUR combined with [CAR + ALP + PIN]. As shown in Appendix A, the results were the same in the presence of a significant synergistic inhibitory effect due to the combination of CUR and [CAR + ALP + PIN]. While 10.51 μg/mL CAR, 6.27 μg/mL ALP, and 15.74 μg/mL PIN combined with 20.24 μg/mL CUR corresponded to 1.90 mg/mL AKH in combination with 20.24 μg/mL CUR, more than 60% ACR was eliminated, which is far beyond the ACR clearance rates of AKH (35.61%) and CUR (11.46%) when they were added alone. Our results showed that when AKH in the range of 1.90–60.80 g/kg and CUR in the range of 0.02–0.65 g/kg are combined with a final concentration ratio of the two of 1:0.01, there is a synergistic inhibitory effect on ACR; namely, the addition of a small amount of CUR greatly improved the ACR inhibition efficiency of [CAR + ALP + PIN] (simulating the AKH composition).

Now that an excellent synergistic inhibition effect on ACR could be achieved by the combination of [CAR + ALP + PIN] and CUR, how about the dosage of the combined inhibitor compared with that of a single one? As listed in Table 1, at 3- and 6-fold reductions in individual doses, the IC_75_ (=75% inhibition rate) of ACR was reached in the model system with the combined incubation of [CAR + ALP + PIN] (AKH) and CUR at an equimolar ratio, and the calculated CI was 0.38, which also indicated a prominent synergistic effect [41]. Furthermore, the CI (0.38) of the quaternary combination was much lower than that of a binary combination of CAR and ALP (CI, 1.26), CAR and PIN (CI, 0.77), and CAR and CUR (CI, 0.61), suggesting that the synergistic effect of the four compounds was stronger than that of the binary combination.

To further understand how the four inhibitors work together quantitatively, the products formed by capturing ACR were quantitatively analyzed by LC–MS/MS, single and combined. Our data showed that the level of ALP-ACR, PIN-ACR, and CUR-ACR adducts produced in the polyphenol combination system was significantly increased compared with the single polyphenol system. As illustrated in Table 2, up to 170.0%, 338.1%, 10.9%, and 5.4% of the ACR adducts of ALP, CUR, PIN, and CAR were increased, respectively, compared with single use. Structurally, for ALP, CAR can be transformed into ALP by closing the C-ring to form more ALP-ACR when CAR and ALP are incubated with ACR [36]; for CUR or PIN, the interplay among the four compounds may enable them to capture more ACR in order to form more ACR adducts due to CUR and PIN having two active sites that can bind ACR (Figure 4c,d). Alternatively, as seen above, CAR, in combination with ALP, PIN, or CUR, respectively, has also been proven to have synergistic effects, and can improve the inhibition rate of ACR in binary combination (Figure 2). Thus, we speculated that the quaternary combination particularly promotes the ability of ALP, PIN, and CUR to capture ACR, thus forming more ACR adducts when [CAR + ALP + PIN] and CUR were mixed and incubated with ACR in combination.

### 3.6. Synergistic Inhibitory Effect of AKH and CUR in Combination on ACR in Roasted Pork

Both AKH and CUR are medicinal and edible foods in China that are always used as food additives in meat cooking and processing. According to the National Food Safety Standard GB 2760-2014, the content of CUR, as a seasoning, cannot exceed 0.1 g/kg. According to the calculation results by the Chou–Talalay equation in Appendix A, when 0.10 g/kg CUR and [CAR + ALP + PIN] (equaling around 10.64 g/kg AKH) in combination were added in the model system, the ACR inhibitory rate presumably reached 76.62%.

In order to verify the synergistic inhibitory effect of AKH and CUR during actual food processing, we designed a series of experiments in which AKH and CUR were added as a complex and separately during roasted pork processing. As shown in Figure 5a, about 30.12% of the ACR could be removed with the addition of AKH (group C), which approximated the effect of adding an equimolar of [CAR + ALP + PIN] (30.08%, group B); in addition, there was no significant difference in the inhibition rate of ACR between AKH + CUR (group F) and [CAR + ALP + PIN] + CUR (group E). This result indicated that AKH, as a spice, can be used as a substitute for three flavonoids (CAR, ALP, and PIN) in a ternary combination to inhibit ACR produced in roasted pork processing. Furthermore, when 10.64 g/kg AKH (1%) and 0.1 g/kg CUR (0.01%) were used in combination, the inhibition rate of ACR reached 71.50% in the roasted pork sample, which is close to the result (76.62%) of the model system in Appendix A and much higher than the rates of 30.12 (group C) and 35.65% (group D) that were observed when AKH and CUR, respectively, were used alone. Hence, AKH, as a carrier, can replace [CAR + ALP + PIN] and play a synergistic role in combination with CUR. However, to determine whether only three flavonoids in AKH have synergistic effects with CUR, or whether other components in AKH are involved, further research is needed.

In addition, according to our previous results, 54% ACR removal required the addition of 6% AKH into roasted pork [32]; now, with the addition of only 1% AKH in combination with 0.01% CUR, up to 71.50% of the ACR could be removed in roasted pork. This inhibition rate is very close to that of our previous study in which CUR (0.03%) and quercetin (0.03%) cooperated to capture the ACR (72%) that formed in roasted chicken wings [34]. However, quercetin is not allowed to be used as a food additive in most countries except Japan, and the addition of 0.03% CUR exceeds the national standard (GB 2760-2014), whereas 1.0% AKH and 0.01% CUR are very popular and commonly used in meat processing.

### 3.7. Synergistic Scavenging Mechanism on ACR of AKH and CUR in Combination in Roasted Pork

ACR adducts that conjugated with CAR, ALP, and PIN from AKH and CUR, such as CAR-ACR, ALP-ACR, PIN-ACR-1, PIN-ACR-2 (Figure 5b-B), CUR-ACR-1, and CUR-ACR-2 (Figure 5b-C), were also detected in roasted pork with AKH and CUR added by using LC–MS/MS, which is similar to the result of the simulated model system (Figure 3). This result emphasized that CAR, ALP, and PIN from AKH, together with CUR could capture the ACR produced in roasted pork to form their ACR adducts, thus eliminating ACR. Our data indicated that this was one of the important pathways to reduce the level of ACR during pork roasting, especially after the generation of ACR. Undeniably, another pathway to prevent the formation of ACR may also be through the antioxidant abilities of three flavonoids of AKH and CUR.

With the new concerns, will the generated adducts cause safety issues? Increasingly, studies have reported that dietary polyphenols could capture ACR to form the corresponding adducts [10,28,29,30,31]. However, the safety of these ACR polyphenol adducts has only been mentioned in a few available reports [42,43]; for example, the cytotoxicity of the ACR adducts of rutin against Caco-2 and GES-1 cells was significantly reduced compared with ACR.

We speculate that the toxicity of the four polyphenol adducts is less than that of ACR because the active carbonyl group of ACR forms an adduct with polyphenols, resulting in a lower reactivity of the adduct than ACR. However, this needs to be confirmed through systematic research, especially *in vivo* experiments, which is the subject of future research. In addition, in our previous study, ACR adducts of CAR, ALP, and PIN were detected in mouse urine and fecal samples after feeding them CAR or ALP [36], which indicated that these adducts can be excreted through feces or urine rather than accumulating in the body, and which also indicated that these adducts cannot be reversed during digestion.

## 4. Conclusions

For our study, we selected the remarkably high-efficiency ACR inhibitors from natural polyphenols in spices. After obtaining their IC_50_ values for trapping ACR, we analyzed binary combinations of polyphenols using the Chou–Talalay method and determined that the compounds ALP, PIN, and CUR had a synergistic inhibitory effect on ACR in combination with CAR. We further demonstrated that the quaternary combination of CUR with CAR, ALP, and PIN (with 6-fold of CUR to 3-fold of [CAR + ALP + PIN] reduction in a single dose) could synergistically trap ACR by forming more ACR adducts of the four polyphenols using LC–MS/MS. Then, the dosage of AKH was found out under the condition of a fixed proportion of the three compounds, ensuring the best synergistic inhibition rate. On this basis, after complexation with CUR (0.01%), a much lower addition percentage of AKH (1%), as a carrier of CAR, ALP, and PIN, to roasted pork could achieve a better ACR inhibition effect, and the LC–MS/MS data suggested that the adducts of ACR conjugated with polyphenols, such as CAR-ACR, ALP-ACR, PIN-ACR-1, PIN-ACR-2, CUR-ACR-1, and CUR-ACR-2, were all present in the roasted pork samples. Our findings not only provide a strategy for inhibiting ACR through combined spice formulas, but also demonstrate the synergistic inhibitory effects of some polyphenols from food spices on ACR by binding ACR to form adducts during roasted pork processing. Further study is needed to explore the ability of polyphenols to eliminate ACR synergistically *in vivo*.

## Figures and Tables

**Figure 1 foods-12-02326-f001:**
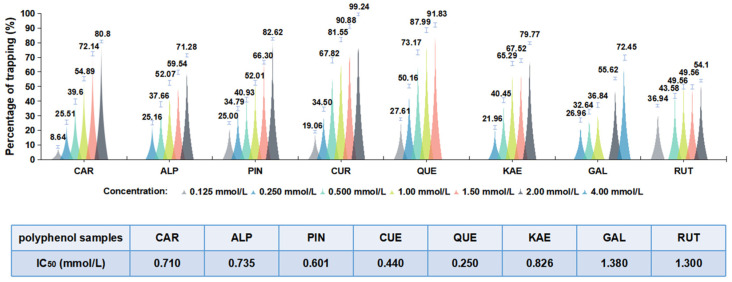
The inhibition rates and IC_50_ values of 0.5 mmol/L CAR, ALP, PIN, CUR, QUE, KAE, GAL, and RUT on ACR (0.5 mmol/L) incubated in a molar ratio of 1:1 at 100 °C for 60 min in PBS (0.1 mol/L, pH 7.0).

**Figure 2 foods-12-02326-f002:**
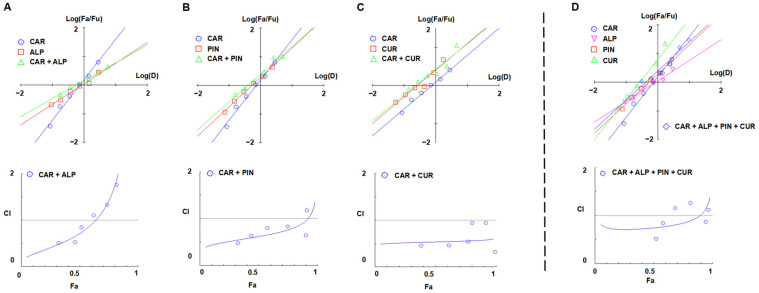
Effects of ACR (0.5 mmol/L) incubated with 0.125, 0.25, 0.5, 1.0, 2.0, and 4.0 times the IC_50_ of each compound: (**A**) CAR and ALP, (**B**) CAR and PIN, and (**C**) CAR and CUR, or (**D**) ALP, CAR, PIN, and CUR individually or in combination at a molar ratio of 1:1 at 100 °C for 60 min in PBS (0.1 mol/L, pH 7.0). The interactions on inhibitory effects were analyzed using the Chou–Talalay method, and the combination index (CI) theorem offers a quantitative definition for the additive (CI = 1), synergistic (CI < 1), or antagonistic (CI > 1) effect of an inhibitor combination. [Different uppercase letters indicate significant (*p* < 0.05) differences in the inhibitors; different lowercase letters indicate significant (*p* < 0.05) differences in the samples among the concentrations].

**Figure 3 foods-12-02326-f003:**
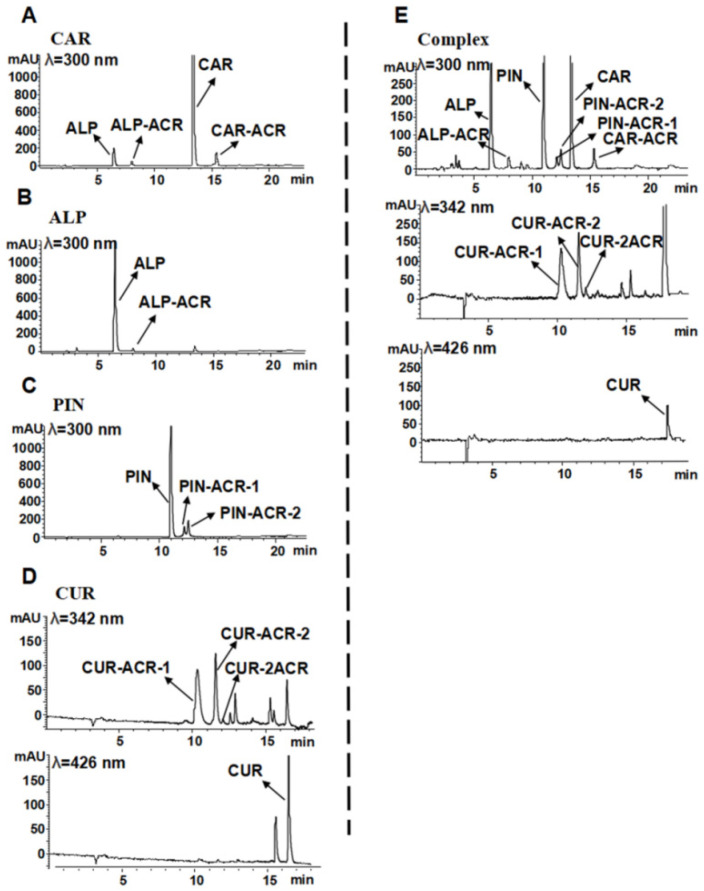
LC–MS/MS chromatograms of ACR (0.5 mmol/L) incubated with 0.125, 0.25, 0.5, 1.0, 2.0, and 4.0 times the IC_50_ of (**A**) ALP, (**B**) CAR, (**C**) PIN, and (**D**) CUR individually or (**E**) in quaternary combination at 100 °C for 60 min in PBS (0.1 mol/L, pH 7.0).

**Figure 4 foods-12-02326-f004:**
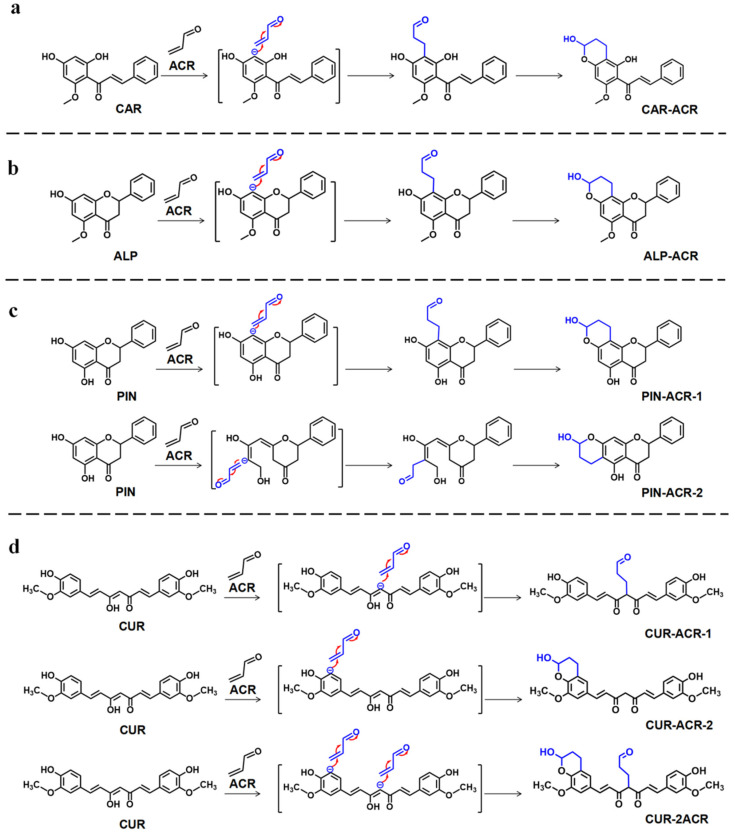
The reaction pathway of (**a**) CAR, (**b**) ALP, (**c**) PIN, and (**d**) CUR capturing ACR to form adducts.

**Figure 5 foods-12-02326-f005:**
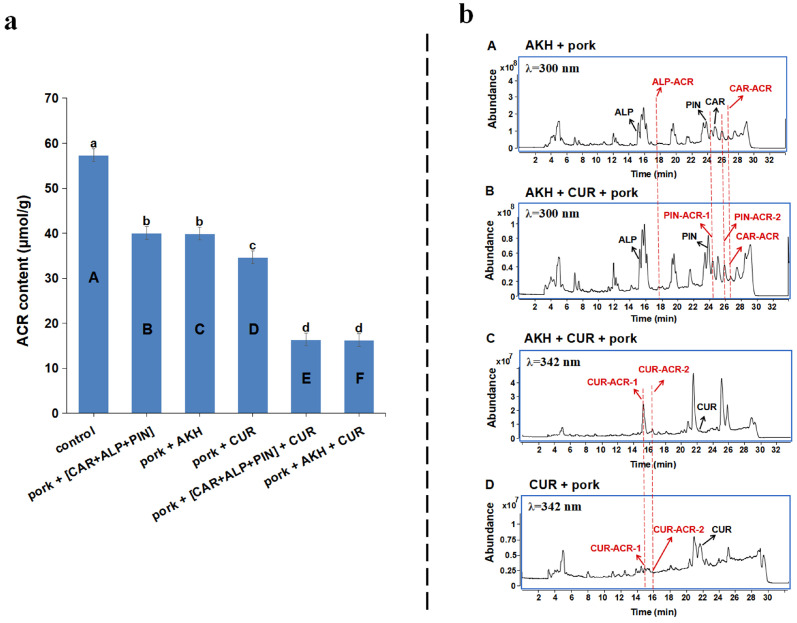
(**a**) The inhibitory effects of AKH and CUR individually and in combination on ACR in roasted pork. Group A: control, group B: with the complex of [CAR + ALP + PIN] according to the fixed proportion of the three compounds in 1.0% AKH, group C: with AKH (1.0%), group D: with CUR (0.1%), group E: with [CAR + ALP + PIN] (1.0% AKH) + CUR (0.1%), and group F: with AKH (1.0%) + CUR (0.1%) [Different lowercase letters (a–d) indicate significant (*p* < 0.05) differences in the samples.]; (**b**) LC–MS/MS chromatograms of AKH (**A**) and CUR (**D**) individually and in combination (**B**,**C**) on ACR in roasted pork.

**Table 1 foods-12-02326-t001:** Effect of [CAR + ALP + PIN] according to the fixed proportion of the three compounds in AKH and CUR on ACR, in combination versus individually.

Compound	IC_50_			IC_75_	IC_75_	IC_75_
	Monotherapy (mmol/L)	Combination (mmol/L)	CI	Monotherapy (mmol/L)	Combination (mmol/L)	CI
[CAR + ALP + PIN]	0.169	0.047	0.383	0.764	0.184	0.386
CUR	0.379	0.041		1.119	0.162	

**Table 2 foods-12-02326-t002:** The amount of the ACR adducts of AKH and CUR incubated with ACR, in combination versus individually.

Compound	Adduct	Single Content (mg/L)	Complex Content (mg/L)	Increase Rate (%)
CAR	CAR-ACR	41.3 ± 10.2 ^ab^	43.5 ± 12.4 ^bc^	5.4 ± 1.2 ^d^
ALP	ALP-ACR	14.9 ± 7.1 ^c^	40.2 ± 8.9 ^bc^	170.0 ± 23.4 ^b^
PIN	PIN-ACR-1	28.1 ± 3.9 ^b^	31.1 ± 1.7 ^c^	10.9 ± 4.2 ^d^
	PIN-ACR-2	51.1 ± 13.9 ^a^	55.2 ± 11.8 ^bc^	8.1 ± 2.5 ^d^
CUR	CUR-ACR-1	56.6 ± 1.6 ^a^	96.9 ± 10.6 ^a^	71.3 ± 13.4 ^c^
	CUR-ACR-2	41.7 ± 15.2 ^ab^	70.9 ± 33.6 ^ab^	70.2 ± 13.5 ^c^
	CUR-2ACR	2.9 ± 0.4 ^d^	12.8 ± 2.1 ^d^	338.1 ± 7.4 ^a^

Note: ^a–d^ different letters in the same row represent significant differences (*p* < 0.05).

## Data Availability

The data presented in this study are available on request from the corresponding authors. The data are not publicly available, due to the request for funding scientific research projects.

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
