# Peer review of "Synergistic Inhibitory Effect of Multiple Polyphenols from Spice on Acrolein during High-Temperature Processing"

_foods, 2023, doi:10.3390/foods12122326_

Round 1
Reviewer 1 Report
Acrolein is a cell-toxic aldehyde and increases the incidence of a variety of chronic diseases, including cardiovascular disease, diabetes mellitus, cancer, and Alzheimer's disease. Food is one of its main This paper aims to study the synergy that may exist between a variety of polyphenols to be included in food (through a spice) that have already demonstrated the ability to reduce the effect of acrolein. Using the hou-Talay method the synergistic inhibitory effect of binary, ternary and quaternary combinations of polyphenols (mixed with a spice) on acrolein is studied. Finally, a spice recipe was developed with a rich flavor to food to be used as a new and promising acrolein inhibitor in food processing. The results obtained suggested that selective complex polyphenols can synergistically remove this toxic that occurs in its processing.
Nowadays it is very interesting to investigate new strategies to reduce food products or metabolites that can cause or enhance problems in the consumer's health. Contributing knowledge by studying strategies to avoid or reduce it is necessary and of great interest to other scientists, wich is the purpose of the present work.
I consider that the following revisions should be made:
Materials and methods. Point 2.2.- It is not clear how the solutions are carried out (i.e. solvents used) of the different polyphenols (CAR, ALP, PIN, CUR, QUE, KAE and RUT). Please explain clearly.
It is very confusing to follow points 3.6 and 3.7. For a better understanding of the explanation, it is advisable to accompany it with a visual, make a flowchart or by means of a table.
The work remains incomplete, since it is necessary to test the formula, combination of polyphenols with the spice, that presents an inhibitory effect on acrolein. This can be done as a first approximation in a simple way using the formula in a piece of pork and doing a organoleptic analysis among the team members themselves. Since it does not make sense to continue studying if there is no organoleptic viability.
Reviewer 2 Report
Manuscript ID: foods-2408414, titled “Synergistic inhibitory effect of multiple polyphenols from spice on acrolein during high-temperature processing” cites 42 references, which are in line with the theoretical basis of the experiment and justify and techniques used in the study. The quality of the scientific English language is outstanding; the text is logically structured.
I have the following specific comments:
The authors mentioned TDI value for acrolein in the Introduction regardless organism on which it has been established, it perhaps would be better to check NOEL value.
The main concern is about quantitative analysis and data on the amounts of CAR, ALP, and PIN in AKH; whether the authors refer to a previous trial that reported these important results for the experiments in this study or is it integrative part of current study (lines: 140-141; 378-379). In both cases, it would be good to provide a chromatogram and results on the content of these three polyphenols in Alpinia katsumadai Hayata (AKH).
When CAR, ALP, PIN, and CUR were mixed and reacted with ACR, numerous ACR adducts, (CAR-ACR, ALP-ACR, PIN-ACR-1, PIN-ACR-2, CUR-ACR-1, CUR-ACR-2, and CUR-2ACR) were detected. Is there possibility to make any relationship between number of ACR adducts per compounds with their IC50 values?
Line 53: cardamonin (CAR) AND alpinetin (ALP)
Line 73: abbreviation for CI is defined letter, please do it on the first appearance in the text.
Reviewer 3 Report
Dear Authors,
The work is really very interesting, however some modifications are necessary to make it easier to read.
Keywords: I would recommend not putting the same keywords that are already in the title
Introduction
Line 41: To insert a reference.
Line 48: to insert some examples of unauthorized polyphenols.
Materials and methods
The materials and methods are really very long.
Line 96-98: he could be included in the acknowledgments.
Line 110-111: the equations must be presented following the guidelines of the journal.
Line 163-164: more than equation of the calibration curve, it would be interesting to report the range of linearity of the calibration curve.
Line 183: you worked on the fresh sample, therefore not freeze-dried. Right?
Results and Discussion
The scavenging capability has only been tested at pH 7. Haven't you thought about using other pH as well?
Round 2
Reviewer 3 Report
Dear Author,
thanks for editing the paper.
Author Response
Dear reviewer,
Thank you for reviewing this article and making valuable suggestions for modification.